The polarization of microglia and infiltrated macrophages in the injured mice spinal cords: a dynamic analysis

Li Jing-Lu 1 2
Fu Gui-Qiang 1 2
Wang Yang-Yang 1 2
Bian Ming-Ming 1 2
Xu Yao-Mei 1 2
Zhang Lin 1 2
Chen Yu-Qing 1 2
Zhang Nan 1 2
Ding Shu-Qin 1 2
Wang Rui 2
Fang Rui 3
Tang Jie 2
Hu Jian-Guo 1 2 jghu9200@163.com
http://orcid.org/0000-0002-3889-835X Lü He-Zuo 1 2 lhz233003@163.com
1 Clinical Laboratory, The First Affiliated Hospital of Bengbu Medical College , Bengbu , China
2 Anhui Key Laboratory of Tissue Transplantation, The First Affiliated Hospital of Bengbu Medical College , Bengbu , China
3 Department of Clinical Medical, Bengbu Medical College , Bengbu , China
Constantin Gabriela
Electronic publication date: 2023 Feb 21
Publication date: 2023
Volume: 11
Electronic Location ID: e14929
Received 2022 Oct 21; Accepted 2023 Jan 30
Copyright: © 2023 Li et al.
Copyright year: 2023
Copyright holder: Li et al.
License: This is an open access article distributed under the terms of the Creative Commons Attribution License, which permits unrestricted use, distribution, reproduction and adaptation in any medium and for any purpose provided that it is properly attributed. For attribution, the original author(s), title, publication source (PeerJ) and either DOI or URL of the article must be cited.
License URL: https://creativecommons.org/licenses/by/4.0/

Keywords: Spinal cord injury, Microglia, Macrophages, Dynamic analysis, Time window

Funding: National Natural Science Foundation of China 82072416 and 81772321 512 Talent Cultivation Plan of Bengbu Medical College 51201109 First Affiliated Hospital of Bengbu Medical College BYYFY2022TD001 This study was supported by grants from the National Natural Science Foundation of China (82072416 and 81772321), the 512 Talent Cultivation Plan of Bengbu Medical College (51201109), and the high level scientific and technological innovation team fund of the First Affiliated Hospital of Bengbu Medical College (BYYFY2022TD001). The funders had no role in study design, data collection and analysis, decision to publish, or preparation of the manuscript.

==============================
Background

Following spinal cord injury (SCI), a large number of peripheral monocytes infiltrate into the lesion area and differentiate into macrophages (Mø). These monocyte-derived Mø are very difficult to distinguish from the local activated microglia (MG). Therefore, the term Mø/MG are often used to define the infiltrated Mø and/or activated MG. It has been recognized that pro-inflammatory M1-type Mø/MG play “bad” roles in the SCI pathology. Our recent research showed that local M1 cells are mainly CD45−/lowCD68+CD11b+ in the subacute stage of SCI. Thus, we speculated that the M1 cells in injured spinal cords mainly derived from MG rather than infiltrating Mø. So far, their dynamics following SCI are not yet entirely clear.

Methods

Female C57BL/6 mice were used to establish SCI model, using an Infinite Horizon impactor with a 1.3 mm diameter rod and a 50 Kdynes force. Sham-operated (sham) mice only underwent laminectomy without contusion. Flow cytometry and immunohistofluorescence were combined to analyze the dynamic changes of polarized Mø and MG in the acute (1 day), subacute (3, 7 and 14 days) and chronic (21 and 28 days) phases of SCI.

Results

The total Mø/MG gradually increased and peaked at 7 days post-injury (dpi), and maintained at high levels 14, 21 and 28 dpi. Most of the Mø/MG were activated, and the Mø increased significantly at 1 and 3 dpi. However, with the pathological process, activated MG increased nearly to 90% at 7, 14, 21 and 28 dpi. Both M1 and M2 Mø were increased significantly at 1 and 3 dpi. However, they decreased to very low levels from 7 to 28 dpi. On the contrary, the M2-type MG decreased significantly following SCI and maintained at a low level during the pathological process.

Introduction

Spinal cord injury (SCI) is a serious neurological disorder, caused by traffic accidents, trauma and other factors (Attal, 2021; Perrouin-Verbe et al., 2021; Quadri et al., 2020). As the bridge of brain and peripheral nerves, the nerve fibers of spinal cord are distributed to the skin, muscles and various internal organs. Once damaged, it will cause serious pathophysiological dysfunctions. For patients, SCI may cause serious physical and mental suffering. The disease also causes serious economic and social burdens (Chay & Kirshblum, 2020). Therefore, to find effective treatment strategies, its pathological mechanism should be deeply explored.

The pathological process of SCI includes primary injury and secondary injury. Primary injury refers to the direct injury of mechanical force to the spinal cord. Secondary injury is triggered by primary injury, which includes local vascular disorder, edema, ischemia, free radical reaction, electrolyte change, inflammation, axon necrosis and demyelination, fibroglial scar and cyst formation (Anjum et al., 2020; Zhang et al., 2021).

Inflammation is one of the important mechanisms of secondary pathological damage of SCI (Mallon, Kwiecien & Karis, 2021). After SCI, with the destruction of blood spinal cord barrier, MG are activated, the inflammatory factors and chemokines are increased, and the peripheral immune cells infiltrate into the injured spinal cord to form an immune microenvironment, resulting in neuronal death and demyelination (Brockie, Hong & Fehlings, 2021; Rezvan et al., 2020; Shields, Haque & Banik, 2020). Following SCI, the different immune cell subsets with different functions affect the local immune microenvironment by producing different cytokines (Mishra et al., 2021). Previous studies have reported that in the injured spinal cord, the cellular components include locally activated MG, infiltrated Mø, lymphocytes, neutrophils, dendritic cells, etc., and these cells are divided into different subsets, some are neuroprotective, while others have neurodamaging effects (Hu et al., 2016; Ma et al., 2015; Milich et al., 2021). Moreover, neurodamaging subsets (e.g., M1, Th1, Th17, etc.) are dominant, which is an important pathological mechanism of SCI (Chen et al., 2021, 2020). However, which of these complex immune cell populations plays a key role? So far, there is still no final conclusion.

Following SCI, a large number of peripheral blood-derived monocytes infiltrate into the injured spinal cord and differentiate into Mø, which are indistinguishable from the local MG, therefore the term Mø/MG was often used to define the infiltrated Mø and/or activated MG in the literature (Fan et al., 2020; Gao et al., 2021; Rismanbaf et al., 2021). However, as research technology advances by leaps and bounds, the peripheral infiltrated Mø and locally activated MG can be identified (Chen et al., 2021, 2020; Milich et al., 2021). Our recent studies have found that proinflammatory M1 cells are absolutely dominant at 7 dpi following SCI, and these cells mainly from MG rather than peripheral infiltrated Mø (Chen et al., 2021, 2020). This suggests that M1 cells derived from MG might be the key inflammatory cells in the immune microenvironment of injured spinal cords. However, the dynamic patterns of MG, infiltrated Mø and their subsets during the whole pathological process of SCI are still unclear. Therefore, the purpose of this study was to explore the dynamic patterns of these cells following SCI using the strategy of combining flow cytometry (FCM) and immunohistofluorescence (IHF).

Materials and Methods

Animals

A total of 150 specific-pathogen free adult female C57BL/6 mice (18–20 g) were obtained from Chang Zhou Cavens Laboratory Animal Ltd. (Chang Zhou, China; license No. SCXK (Su) 2016-0010). Animals were housed as previously described (Chen et al., 2021, 2020). All experimental designs and reports were referred to previous to the previous guidelines (Kilkenny et al., 2011). The surgery protocol was approved by the Animal Care Ethics Committee of Bengbu Medical College. The Animal Ethical Approval number was 2017-037. The mice were randomly divided into sham-operated (sham), 1-, 3-, 7-, 14-, 21- and 28-days post-injury (dpi) groups, using a computer based random order generator (Zhao et al., 2018). The comprehensive description of the total number of mice used is shown in Fig. 1.

Figure 1 Flow chart of the pathway and whole idea of this research.

Contusive SCI

The mice contusive SCI model was established as described previously (Chen et al., 2021, 2020). Specifically, the Infinite Horizon impactor was made by Precision Systems & Instrumentation (Lexington, KY, USA). Anesthetics ( 80 mg/kg ketamine and 10 mg/kg xylazine) for intraperitoneal injection were obtained from Sigma-Aldrich (St. Louis, MO, USA). The T9 spinal cord was impacted with 50 Kdynes force, and the diameter of the impact rod was 1.3 mm. After impact, the spinal cord was filled with blood and edema. Sham-operated mice only underwent laminectomy without contusion. After operation, the animal care and welfare were performed as previously described (Chen et al., 2021, 2020), which included bladder emptying three times per day, relieving pain with meloxicam (5 mg/kg; CSN pharm, Chicago, IL, USA), and preventing infection with chloramphenicol (50 mg/kg; Sangon Biotech, Shanghai, China) for 7 days after surgery. Inclusion criteria: the animals underwent successful contusive SCI, defined by the T9 site filled with blood and edema, and the spinal cord was intact and not ruptured. Exclusion criteria: the degree of injury was not up to standard, postoperative infection or sacrifice.

Flow cytometry

At the indicated time points post-injury, the samples were collected and single-cell suspensions were obtained using Percoll gradient centrifugation as previously described (Chen et al., 2021, 2020). Briefly, the chest cavity was opened with surgical scissors to expose the heart. The ventricle was clamped with a vascular clamp to fix the heart. The No. 7 needle was inserted into the left ventricle. At the same time, a small opening was cut on the right atrium so that the blood and lavage solution could be drained. Then, 10 ml of 0.01 M phosphate-buffered saline (PBS) buffer solution (pH = 7.4) was slowly injected at 250 ml/h with a microinjection pump. After perfusion, the 5 mm spinal cord segments which contained the injury center were taken, and the corresponding spinal cord segments were also obtained from sham group. The spinal cords were put into the 45-m nylon mesh and fully ground with the syringe plunger to obtain single cell suspensions. To obtain enough cells for analysis, three spinal cord segments were mixed for one test. The Percoll gradient centrifugation (Amersham Pharmacia Biotech, Piscataway, NJ, USA) was used to separate the single cells. Table 1 showed the fluorescent labeled antibodies used in this study to identify different immune cell subtypes. To eliminate the background staining caused by the non-specific binding of the antibody, the immunoglobulin with the same species, subtype, dose and fluorescein as the primary antibody was used as the isotype control. The cells were collected using a BD Accuri flow cytometer (Becton Dickinson, San Diego, CA, USA), and the data were analyzed using FlowJo7.6.1 software (FlowJo; LLC, Ashland, OR, USA).

Table 1 Antibodies used in the study.

Antigen	Host species and clone	Cat. # or Lot#	RRID	Conjugation	Source	Used concentration	Methods	
CD11b	Rat monoclonal	14-0112-82	AB_467108	NO	Invitrogen	1:200	IHF	
CD45	Rat monoclonal	14-0451-82	AB_467251	
CD68	Rat monoclonal	MA5-16674	AB_2538168	
Arg1	Rabbit polyclonal	PA5-29645	AB_2547120	
CCR7	Rabbit polyclonal	ab191575		Abcam	
TMEM119	Rat monoclonal	ab209064	AB_2800343	
Rat IgG (H+L)	Goat polyclonal	112-095-143	AB_2338199	Fluorescein (FITC)	Jackson ImmunoResearch	
Rabbit IgG (H+L)	Goat polyclonal	111-025-144	AB_2337932	Rhodamine (TRITC)	
CCR7	Rat monoclonal	47-1971-82	AB_2573974	APC-eFluor 780 (AF780)	Invitrogen	0.25 µg/test	FCM	
IgG2b kappa Isotype Control	Rat	47-4321-82	AB_1271997	
CD11b	Rat monoclonal	12-0112-81	AB_465546	PE	0.125 µg/test	
IgG2b kappa Isotype Control	Rat	12-4031-82	AB_470042	0.25 µg/test	
CD68	Rat monoclonal	MA5-16676	AB_2538170	FITC	
IgG2b kappa Isotype Control	Rat	11-4031-82	AB_470004	
CD45	Rat monoclonal	17-0451-82	AB_469392	APC	0.125 µg/test	
IgG2b kappa Isotype Control	Rat	17-4031-82	AB_470176	

Immunofluorescence double-staining

At the indicated time points post-injury, mice were euthanized and perfused with PBS as described in “Flow cytometry”. Then, the mice were perfused with 10 mL of 4% paraformaldehyde (PFA) at a rate of 180 mL/h. After perfusion, the 5 mm spinal cord segments which contained the injury center were collected and fixed in 10 mL of 4% PFA solution at 4 °C overnight. The next day, the spinal cords were removed from 4% PFA solution and placed in 20% sucrose solution (prepared in PBS) at 4 °C overnight. The third day, the spinal cords were transferred to 30% sucrose at 4 °C, until the samples sinking to the bottom. This process usually needs 1 day. Next, the embedding agent (Tissue-Tek; Sakura Finetek USA Inc., Torrance, CA, USA) was used to embed the spinal cord segments at −20 °C. The 6 μm thick transverse sections were cut using a Leica CM1900 cryostat (Leica Microsystems, Bannockburn, IL, USA). The IHF assay was performed as previously described (Chen et al., 2021, 2020). Briefly, the slides were washed three times with 0.01 M PBS to completely clear the embedding agent. When the slides were left to dry, the blocking solution (0.01 M PBS containing 10% normal goat serum) were used for 2 h at room temperature to eliminate the background staining caused by the non-specific binding of antibodies. After cleaning the blocking solution, the primary antibodies with appropriate concentration were incubated overnight at 4 °C. The next day, the slides were washed three times with 0.01 M PBS to completely remove the unbound antibodies. Then, the secondary antibodies with appropriate concentration were incubated at 37 °C for 1 h. The primary and FITC and RHO-conjugated secondary antibodies were shown in Table 1. After the second antibody incubation, the 0.01 M PBS was used to wash the slides for three times, and the 1 μg/ml Hoechst 33342 (Cat# B2261; Sigma-Aldrich, St. Louis, MI, USA) containing medium was used to coverslip the slides. Finally, the slides were examined using a ZWISS Axio observation microscope (Carl Zeiss, Oberkochen, Germany). The cell quantification was performed as previously described (Chen et al., 2021, 2020). Specifically, for each spinal cord, the cells of 5 complete cross-sections containing the injury epicenter (0 mm), rostral (1 and 0.5 mm) and caudal (−1 and −0.5 mm) were counted.

Statistical analyses

The SPSS software v.14.0 (SPSS Inc., Chicago, IL, USA) was used to statistical analysis. The non-parametric Kruskal Wallis analysis of variance (ANOVA) following by the individual Mann-Whitney U test was used. The P < 0.05 was considered to be statistically significant.

Results

Temporal pattern of MG and infiltrated Mø following SCI: the flow cytometry (FCM) analysis

To determine the temporal pattern of MG and infiltrated Mø, a panel of cell markers (CD11b, CD45 and CD68) was examined by FCM. Here, CD45high population was peripheral infiltrated leukocytes, CD68+CD11b+ population was activated Mø and MG, CD45highCD11b+ population was peripheral infiltrated Mø, CD45−/lowCD11b+ population was MG, CD45highCD68+CD11b+ population was activated peripheral infiltrated Mø, CD45−/lowCD68+CD11b+ population was activated MG, and CD45highCD68−CD11b− population was peripheral infiltrated leukocytes excluding Mø (Fig. 2A).

Figure 2 Temporal pattern of MG and infiltrated Mø following SCI detected by FCM.

(A) The representative pictures of FCM in sham and injured spinal cords. (B–L) The temporal pattern of the indicated cell populations after SCI. Data represent mean ± SD (n = 6). *P < 0.05, **P < 0.01 (non-parametric Kruskal-Wallis ANOVA, following by the individual Mann-Whitney U test).

Figure 2B showed that CD11b+ cells had no significant difference among sham, 1 and 3 dpi groups (P > 0.05, n = 6). However, at 7 dpi, the proportion increased significantly and reached to peak, although decreasing at the later time points (14, 21 and 28 dpi), they remained at high levels comparing with sham, 1 and 3 dpi groups (P < 0.01, n = 6).

Figure 2C showed that CD68+ cells were the lowest in the sham group comparing with the injured groups (P < 0.01, n = 6). The proportions increased significantly after injury, reached to peak at 7 dpi, and maintained at high levels at 14, 21 and 28 dpi. There were no significant differences among 7, 14, 21 and 28 dpi (P > 0.05, n = 6). However, CD68+ cells in these four groups were significant more comparing with sham, 1 and 3 dpi (P < 0.05 or 0.01, n = 6).

Figure 2D showed that CD11b+CD68+ cells were extremely rare in sham group, however, they increased significantly in the injured groups (P < 0.01, n = 6). The proportions had no significant differences between 1 and 3 dpi (P > 0.05, n = 6). However, it reached to peak at 7 dpi, and then decreased, but remained at high levels at 14, 21 and 28 dpi.

Figure 2E showed that CD45high cells were extremely rare in sham group, and they gradually increased after injury, peaked at 7 and 14 dpi, and then decreased, but still maintained at high levels at 21 and 28 dpi.

Figure 2F showed that CD11b+CD45high cells were also extremely rare in sham group, and they significantly increased after injury, peaked at 7 dpi, and then decreased, but still maintained at high levels at 14, 21 and 28 dpi. The proportion of CD11b+CD45high cells in each group is significantly lower than that of their corresponding CD11b+ cells (Fig. 2B).

Figure 2G showed that CD11b+CD45−/low cells had no significant difference among sham, 1 and 3 dpi groups (P > 0.05, n = 6). However, at 7 dpi, the proportion increased significantly and reached a peak, although decreased at the later time points (14, 21 and 28 dpi), they still maintained at high levels comparing with sham, 1 and 3 dpi groups (P < 0.01, n = 6). Except for the 1 and 3 dpi groups, CD11b+CD45−/low cells constitute the majority of CD11b+ cells (Fig. 2B).

Figure 2H showed that CD68+CD45high cells were also extremely rare in sham group, and they rapidly increased after injury. Up to 28 dpi, they still maintained at high levels. The proportion of CD68+CD45high cells in each SCI group is significantly lower than that of their corresponding CD68+ cells (Fig. 2C).

Figure 2I showed that the percentages ofCD68+CD45−/low cells in sham group was the lowest comparing with the injured groups (P < 0.05 or 0.01, n = 6). The proportions had no significant differences at 1 and 3 dpi (P > 0.05, n = 6). However, they reached to peak at 7 and 14 dpi, and remained at high levels at 21 and 28 dpi. The proportion of CD68+CD45−/low cells in each SCI group is significantly lower than that of their corresponding CD68+ cells (Fig. 2C).

Figure 2K showed that CD68+CD11b+CD45high cells were extremely rare in sham group, and they rapidly increased after injury, peaked at 1, 3 and 7 dpi. Compared with the sham group, these cells in the 1, 3 and 7 dpi groups were significantly more (P < 0.01, n = 6). Although, comparing with 1, 3 and 7 dpi groups, the proportions decreased to the lower levels at the later time points (14, 21 and 28 dpi) (P < 0.05, n = 6), they still maintained at higher levels comparing to the sham group (P < 0.05, n = 6). Comparing with Fig. 2H, the proportion of cells in Fig. 2K is lower. This indicated that the activated peripheral infiltrated Mø in the injured spinal cord are significantly inferior to activated MG.

Figure 2L showed that CD68+CD11b+CD45−/low cells were also extremely rare in the sham group, and they gradually increased after injury, peaked at 7 dpi, and then decreased, but still maintained at high levels at 14, 21 and 28 dpi. Comparing to sham group, the percentages of CD68+CD11b+CD45−/low cells in all SCI groups had significant differences (P < 0.01, n = 6).

Temporal pattern of MG and infiltrated Mø following SCI: the immunohistofluorescence (IHF) analysis

To verify the temporal pattern of MG and infiltrated Mø detected by FCM, the spinal cords from several representative time points after SCI (sham, 1, 7 and 28 dpi) were selected for IHF analysis. CD11b, CD68 and TMEM119 antibodies were used for immunofluorescence labeling (Fig. 3). Here, TMEM119+CD11b+ cells are total MG, TMEM119−CD11b+ cells are monocyte-derived Mø, TMEM119+CD68+ cells are activated MG, TMEM119−CD68+ cells are activated monocyte-derived Mø, respectively (Figs. 3A–3H).

Figure 3 Temporal pattern of MG and infiltrated Mø following SCI detected by IHF.

(A–D) The representative pictures of TMEM119 (green) and CD11b (magenta) in the spinal cords of sham and contusion epicentre at T9 segmental level (A: sham; B: 1 dpi; C: 7 dpi; D: 28 dpi). (E and F) Quantitative analysis the cells of CD11b+TMEM119+ (E) and CD11b+TMEM119− (F). Data represent mean ± SD (n = 6). *P < 0.05, **P < 0.01. (non-parametric Kruskal-Wallis ANOVA, following by the individual Mann-Whitney U test).

The representative images showed that TMEM119+CD11b+ cells could be detected in all groups (Figs. 3A–3D). The statistical results (Fig. 3E) showed that the number (cells/mm2) of TMEM119+CD11b+ cells had no significant difference between sham (117.50 ± 19.30) and 1 dpi (200.33 ± 16.59) groups (P > 0.05, n = 6). Comparing with the other three groups, there were more TMEM119+CD11b+ cells in the 7 dpi (537.33 ± 99.80) (P < 0.01, n = 6). Although, the cells were decreased at 28 dpi (308.33 ± 50.27), the number still significantly more comparing with sham and 1 dpi groups (P < 0.01, n = 6). In the sham group, TMEM119−CD11b+ cells were extremely rare (Fig. 3A), and they significantly increased in the three SCI groups (Fig. 3B-D). The statistical results (Fig. 3F) showed that the numbers of TMEM119−CD11b+ cells in all three SCI groups were significant more than that of sham (70.17 ± 7.65) group (P < 0.01, n = 6). Comparing with the other three groups, there were also more TMEM119−CD11b+ cells in 7 dpi (201.80 ± 42.53) group (P < 0.05, n = 6).

In Fig. 4A, both TMEM119+CD68+ and TMEM119−CD68+ cells were extremely rare in sham group. However, both of them could be detected in all SCI groups (Figs. 4B–4D). The statistical results (Figs. 4E and 4F) showed that the numbers of these two types of cells had significant differences among sham and SCI groups (P < 0.01, n = 6). Compared with the three other groups, there were most TMEM119+CD68+ cells in 7 dpi (473.50 ± 64.48) group (P < 0.01, n = 6). Although, the number of these cells decreased at 28 dpi (269.67 ± 42.49), it still significantly more comparing with sham (6.67 ± 6.31) and 1 dpi (156 ± 43.75) groups (P < 0.01, n = 6). Comparing with the other two groups, TMEM119−CD68+ cells were most in 1 (155.00 ± 18.51) and 7 dpi (124.75 ± 34.88) groups (P < 0.01, n = 6).

Figure 4 Temporal pattern of activated MG and infiltrated Mø following SCI detected by IHF.

(A–D) The representative pictures of TMEM119 (green) and CD68 (magenta) in the spinal cords of sham and contusion epicentre at T9 segmental level (A: sham; B: 1 dpi; C: 7 dpi; D: 28 dpi). (E and F) Cellular quantitation of CD68+TMEM119+ (E) and CD68+TMEM119− (F). Data represent mean ± SD (n = 6). *P < 0.05, **P < 0.01. (non-parametric Kruskal-Wallis ANOVA, following by the individual Mann-Whitney U test).

Temporal pattern of SCI-induced M1 and M2 differentiation of Mø and MG: the FCM analysis

To further explore the temporal pattern of SCI-induced M1 and M2 differentiation of Mø and MG, FCM was used by combining CD68, CD45, CD11b and CCR7 antibodies.

As shown in Fig. 5A, the same size “region” of total CD11b+ cells (R1) were set for each sample in the pseudocolor plots of CD45/CD11b, and then the percentage of each cell population was analyzed in the pseudocolor plots of CD68/CCR7 by setting the boundary between negative and positive with isotype-matched antibodies. The statistical results (Fig. 5B) showed that the percentage of CD11b+CD68+CCR7+ M1 cells in the sham group was the lowest. The proportions significantly increased after injury, reached to peak from 7 dpi, and maintained at high levels at 14, 21 and 28 dpi. There were no significant differences among 7, 14, 21 and 28 dpi (P > 0.05, n = 6). However, the percentages in these four groups were significant higher comparing with sham and 1 dpi groups (P < 0.01, n = 6).

Figure 5 Temporal pattern of SCI-induced M1 and M2 differentiation of Mø and MG detected by FCM.

(A, E and I) Representative images of total M1 and M2 cells (A), M1 and M2 MG (E), M1 and M2 Mø (I) detected by FCM in sham and injured spinal cords. (B–D, F–H and J–L) The temporal pattern of the indicated cell populations after SCI. Data represent mean ± SD (n = 6). *P < 0.05, **P < 0.01. (non-parametric Kruskal-Wallis ANOVA, following by the individual Mann-Whitney U test).

Figure 5C showed that CD11b+CD68+CCR7− M2 cells had no significant differences among all groups (P > 0.05, n = 6). However, when converted to ratio (Fig. 5D), the total M1/M2 ratios in all SCI groups were significant increased comparing with the sham group (P < 0.01, n = 6).

As shown in Fig. 5E, in the pseudocolor plots of CD11b/CD45, the same size region of CD11b+CD45−/low cells (R2) were set for each sample, and then the percentages of cell populations were analyzed same as Fig. 5A. The statistical results (Fig. 5F) showed that the percentage of CD11b+CD45−/lowCD68+CCR7+ MG-derived M1 cells in sham group was the lowest comparing with the injured groups, and the differences were statistically significant (P < 0.01, n = 6). The proportions significantly increased after injury, and reached to peak at 14 and 21 dpi, and continued at high level at 28 dpi. The percentages in the groups of 3, 7, 14, 21 and 28 dpi were significantly higher than that of 1 dpi (P < 0.05 or 0.01, n = 6).

Figure 5G showed that CD11b+CD45−/lowCD68+CCR7− MG-derived M2 cells had no significant differences among all groups (P > 0.05, n = 6). However, when converted to ratio (Fig. 5H), the M1/M2 ratio of MG was very low in sham group, and they increased after SCI, peaked at 7 dpi, and then decreased at 21 and 28 dpi. In the group of 7 dpi, the ratio was significantly higher than those of the other groups (P < 0.05, n = 6).

In Fig. 5I, in the pseudocolor plots of CD11b/CD45, the same size region of CD45highCD11b+ cells (R3) were set for each sample, and then the percentages of cell populations were analyzed same as Fig. 5A. Figure 5J showed that the percentages of CD11b+CD45highCD68+CCR7+ peripheral infiltrated M1 cells showed an increasing trend after SCI, and reached the highest levels at 28 dpi. The 7, 14, 21 and 28 dpi groups had statistically significant comparing with sham and 1dpi (P < 0.05 or 0.01, n = 6). Figure 5K showed that the percentages of CD11b+CD45highCD68+CCR7− peripheral infiltrated M2 cells were highest at 3 and 7 dpi, which were statistically significant comparing with the other groups (P < 0.01, n = 6). When converted to ratio (Fig. 5l), the infiltrated M1/M2 ratio was very low in sham group, and there was a transient rise at 1 dpi. Then, the ratios decreased to sham level at 3 and 7 dpi, and then showed an increasing trend from 14 to 28 dpi, it reached the highest levels at 28 dpi. Among sham, 3 and 7 dpi groups, the ratios had no significant differences (P > 0.05, n = 6). However, comparing with the other groups, the ratios were significant lower (P < 0.05 or 0.01, n = 6).

Temporal pattern of SCI-induced M1 and M2 differentiation of Mø and MG: the IHF analysis

In IHF analysis, CD68+CCR7+ and CD68+Arg1+ were used to label total M1 and M2 cells, respectively (Fig. 6); TMEM119+ CCR7+ and TMEM119+Arg1+ cells were M1 and M2 MG, respectively (Figs. 7A–7J). Therefore, (CD68+CCR7+ minus TMEM119+CCR7+) and (CD68+Arg1+ minus TMEM119+Arg1+) were M1 and M2 monocyte-derived Mø, respectively (Figs. 7K and 7L).

Figure 6 Temporal pattern of SCI-induced differentiation of total M1 and M2 cells following SCI detected by IHF.

(A–H) The representative pictures of CCR7 (green) and CD68 (magenta) (A–D), and Arg1 (green) and CD68 (magenta) (E–H) in the spinal cords of sham and contusion epicentre at T9 segmental level (A, E: sham; B, F: 1 dpi; C, G: 7 dpi; D, H: 28 dpi). (I and J) Cellular quantitation of CD68+CCR7+ (I) and CD68+Arg1+ (J). Data represent mean ± SD (n = 6). *P < 0.05, **P < 0.01. (non-parametric Kruskal-Wallis ANOVA, following by the individual Mann-Whitney U test).

Figure 7 Temporal pattern of SCI-induced M1 and M2 differentiation of Mø and MG following SCI detected by IHF.

(A–H) The representative pictures of CCR7 (green) and TMEM119 (magenta) (A–D), and Arg1 (green) and TMEM119 (magenta) (E–H) in the spinal cords of sham and contusion epicentre at T9 segmental level (A, E: sham; B, F: 1 dpi; C, G: 7 dpi; D, H: 28 dpi). (I–L) Cellular quantitation of TMEM119+CCR7+ (I), TMEM119+Arg1+ (J) cells, infiltrated M1 Mø (K) and infiltrated M1 Mø (L). Data represent mean ± SD (n = 6). *P < 0.05, **P < 0.01. (non-parametric Kruskal-Wallis ANOVA, following by the individual Mann-Whitney U test).

The representative images showed that CD68+CCR7+, CD68+Arg1+, TMEM119+CCR7+ and TMEM119+Arg1+ cells were both extremely rare in sham-operated spinal cords (Figs. 6A and 6E, 7A and 7E). However, these cells could be detected in all SCI groups (Figs. 6B–6D, 6F–6H, and 7B–7D, 7F–7H).

The statistical results (Fig. 6I) showed that in the groups of sham, 1, 7 and 28 dpi, the numbers of CD68+CCR7+ cells were 2.67 ± 2.50, 235.33 ± 5.13, 577.17 ± 40.18 and 543.17 ± 31.35, respectively. All SCI groups were significant more than that of sham (P < 0.01, n = 6). Up to 7 dpi, the cell number reached to peak, and continued at high levels at 28 dpi.

In Fig. 6J, the numbers of CD68+Arg1+ cells in the groups of sham, 1, 7 and 28 dpi were 3.83 ± 3.43, 105.33 ± 9.56, 72.83 ± 10.55 and 150.17 ± 22.21, respectively. Although, CD68+Arg1+ cells in all SCI groups were significant more than that of sham group (P < 0.01, n = 6), the 7 dpi had the least number of cells among the three SCI groups, and there were most cells in 28 dpi group.

In Fig. 7I, TMEM119+CCR7+ cells in the groups of sham, 1, 7 and 28 dpi were 3.17 ± 1.83, 205.17 ± 9.97, 412.33 ± 18.04 and 410.17 ± 50.92, respectively. The overall change trend was similar to that of CD68+CCR7+ cells. Figure 7J showed that TMEM119+Arg1+ cells in the groups of sham, 1, 7 and 28 dpi were 4.33 ± 2.42, 85.83 ± 10.68, 50.00 ± 4.29 and 125.00 ± 13.33, respectively. The overall trend was also similar to that of CD68+Arg1+ cells.

In Fig. 7K, the infiltrated CD68+CCR7+ cells in the groups of sham, 1, 7 and 28 dpi were 0.00 ± 0.00, 30.17 ± 11.72, 164.83 ± 45.64 and 133.00 ± 56.67, respectively. All SCI groups were significant more than that of sham group (P < 0.01, n = 6). Up to 7 dpi, the cell number reached to peak. Although the number of cells had a decreasing trend, it remained at a higher level at 28 dpi. Figure 7L showed that although the infiltrated CD68+Arg1+ cells could be detected in SCI groups, they were very rare. The numbers in the groups of sham, 1, 7 and 28 dpi were 0.00 ± 0.00, 19.50 ± 12.65, 22.83 ± 12.04 and 25.17 ± 15.74, respectively. All three SCI groups were significant more than that of sham group (P < 0.01, n = 6).

Summary of dynamic changes of MG, infiltrated Mø and their subsets

As shown in the first row of Fig. 8, the proportions of total Mø/MG in the groups of SCI were gradually increased and peaked at 7 dpi. Although, decreasing at 14, 21 and 28 dpi, they were still maintained at high levels compared with sham, 1 and 3 dpi groups.

Figure 8 Summary of dynamic changes of MG, infiltrated Mø and their subsets.

This is an integrated analysis of Figs. 2–7. The first to seventh lines indicate sham, 1, 3, 7, 14, 21 and 28 dpi, respectively. The first row shows the proportions of total Mø/MG and the other cells. The second row shows the proportions of activated and resting cells in total Mø/MG. The third row shows the proportions of activated Mø and activated MG in the activated Mø/MG. The fourth row shows the proportion of M1 and M2 subsets in the activated MG and infiltrated Mø.

The second row of Fig. 8 showed that most of the Mø/MG were activated following SCI. The third row of Fig. 8 showed that MG were absolutely dominant and Mø were few in the sham-operated spinal cords. In the acute phase of SCI (1 and 3 dpi), the proportions of activated Mø increased significantly. However, with the progression of the SCI, the proportions of activated MG increased nearly to 90% in subacute phase (7 and 14 dpi) and chronic phase (21 and 28 dpi).

Finally, we further analyzed the proportion of M1 and M2 subsets in the activated MG and infiltrated Mø. The fourth row 4 of Fig. 8 showed that in sham, 1, 3, 7, 14, 21 and 28 dpi groups, the proportions of M1 Mø were 7%, 43%, 51%, 7%, 9%, 17% and 12%, respectively, while M2 Mø were 6%, 6%, 23%, 3%, 1%, 1% and 0%, respectively. The proportions of M1 MG were 31%, 43%, 20%, 69%, 79%, 66% and 68%, respectively, and M2 MG were 56%, 8%, 6%, 21%, 11%, 16% and 20%, respectively. These results showed that there are very few peripheral Mø in the sham-operated spinal cords, while the proportion of MG is absolutely dominant, and the MG are mainly M2 subtype. In the acute phase of SCI, the proportion of peripheral infiltrated Mø increased transiently, and M1 Mø are absolutely dominant, but in the subacute phase, M1 MG were absolutely dominant and continued to the chronic phase (28 dpi, the longest time point observed in this study).

Discussion

Following SCI, the local microenvironment of spinal cord is destroyed (Anjum et al., 2020; Fan, Wei & Feng, 2022; Fan et al., 2018). SCI not only leads to neuronal and oligodendrocyte necrosis and astrocyte activation, it also triggers the immune response, which includes the local MG activation and the peripheral immune cell infiltration (Donnelly & Popovich, 2008; Lee et al., 2009). Among these immune cells, some are neuroprotective, while others are destructive. The final outcomes of SCI depend on the dynamic balance of these cells (DiSabato, Quan & Godbout, 2016; Wolf et al., 2002).

Previous studies found that local pro-inflammatory M1-type MG (MG) and/or infiltrated M1-type Mø are absolutely dominant following SCI (Fan et al., 2019; Sato et al., 2012). These suggest that M1 cells may be the key factor for the imbalance of local immune microenvironment of SCI. However, it remains controversial that these M1 cells are mainly from MG or the infiltrated Mø. Our recent research showed that the local M1 cells are mainly CD45−/lowCD68+CD11b+, rather than CD45highCD68+CD11b+ cells in the subacute stage of SCI (Chen et al., 2021, 2020). In fact, CD45 is not only a common marker of peripheral leukocytes, it can also be expressed at a low level in MG (Sedgwick et al., 1998), CD11b is mainly expressed in MG and monocyte-derived Mø (Martin et al., 2017), and CD68 is the common marker of activated MG and Mø (Chen et al., 2015; Greaves & Gordon, 2002). Therefore, CD45highCD68+CD11b+ population was activated peripheral infiltrated Mø, CD45−/lowCD68+CD11b+ population was activated MG, and CD45highCD68−CD11b− population was peripheral infiltrated leukocytes excluding Mø. Thus, we speculated that the M1 cells in the injured spinal cords mainly derived from MG rather than infiltrating Mø. To test this hypothesis, in this study, the mouse SCI model was established, the activation and proportion of MG and infiltrated Mø at different time points following SCI were dynamically observed by a panel of specific cell markers using FCM and IHF.

Generally, in the rodent model, the pathological process of traumatic SCI is divided into the acute (<48 h), subacute (2 to 14 dpi) and chronic (>14 dpi) phases (Rodrigues, Moura-Neto & de Sampaio E Spohr, 2018; Shi et al., 2017). Therefore, in this study, we selected the sham-operated spinal cord as the normal control, 1 dpi as acute, 3, 7 and 14 dpi as subacute, 21 and 28 dpi as chronic phases, respectively.

The results showed the proportions of total Mø/MG peaked at 7 dpi. Although, the proportions decreased at 14, 21 and 28 dpi, they were still maintained at high levels comparing with sham, 1 and 3 dpi. This change trend is consistent with the previous reports (Chen et al., 2015; Kigerl et al., 2009; Wang et al., 2015). We also found that the activated Mø increased significantly at 1 and 3 dpi following SCI. However, with the progression of pathological process, the activated MG dominated absolutely and increased nearly to 90% at 7, 14, 21 and 28 dpi. These demonstrate that in the early stage of SCI, the peripheral Mø infiltrate into the injured area rapidly, and the activated MG and Mø are in a roughly balanced state. This is consistent with previous report (Hellenbrand et al., 2021).

Here, we can find that the percentages of CD68+ cells in 7, 14, 21 and 28 dpi groups were significant higher comparing with sham, 1 and 3 dpi groups, and there were no significant differences among 7, 14, 21 and 28 dpi groups. These may because in the later stage of subacute phase and chronic phase, most of the CD68+ cells are activated MG and Mø, and their number has reached the maximum. The possible reason is that lesion-associated factors (e.g., proinflammatory cytokines, oxygen tension, chemokines, etc.) persist indefinitely in the local microenvironment, and the recruitment of MG/Mø might promote new blood vessel growth and extracellular matrix deposition in these stages of SCI (Kigerl et al., 2009).

In this report, the activated MG are dominant at 7 dpi. Although, this is not consistent with previous report (Hellenbrand et al., 2021), it is consistent with the recent reports which using single-cell RNA sequencing to analyze the temporal changes at molecular and cellular levels in the injured mouse spinal cords (Li et al., 2022; Milich et al., 2021). The possible reason is that MG response following acute SCI limits infiltrated Mø dispersion (Plemel et al., 2020). Accordingly, we further found that both the proportions of M1 and M2 Mø were increased significantly at 1 and 3 dpi. However, they decreased to very low levels from 7 to 28 dpi. This phenomenon also shows that the activated MG might inhibit both infiltrated M1 and M2 Mø. Following SCI, the M1 MG increased and maintained at high levels from 7 to 28 dpi. On the contrary, the proportion of M2 MG decreased significantly after SCI and remained at a low level during the whole pathological process. Based on these results, we can infer that the activated MG are mainly M1 subtype following SCI. They inhibit not only the infiltration of peripheral monocytes, but also the polarization of these cells into M2 Mø. In the same way they can also inhibit the polarization of themselves into M2 MG. Therefore, M1 cells derived from MG are the key cells involved in proinflammatory response of SCI.

Following SCI, the main effector cells are the peripheral infiltrated Mø and resident MG (David & Kroner, 2011). In fact, whether the immune response of these cells is good or bad depends on their subtypes and functional characteristics. Based on their functions, Mø/MG can be divided into M1 and M2 subtypes. M1 cells can damage nerve cells by secreting inflammatory cytokines, while M2 cells can regulate immune inflammatory response, remove necrotic tissue fragments, promote vascular regeneration, tissue reconstruction and repair (Kigerl et al., 2009; Kroner et al., 2014; Wang et al., 2015). In this study, we demonstrated that most Mø and MG in the injured spinal cords are M1 cells, only a small number showing M2 phenotype and they are transient. This shows that the predominance of M1 macroglia and lower number of M2 macroglia and/or Mø may contribute to the early inflammatory response and secondary damage following SCI. Therefore, for clinical transformation and application of Mø/MG, it is very necessary to determine the appropriate time window of these cells for immune intervention. Our temporal dynamic analysis suggests that the acute and early stage of subacute phase may be the window period for immune intervention targeting MG. During this time window, using effective intervention measures to timely inhibit the differentiation of MG into M1 subtype, rather than focusing on the infiltration and activation of peripheral monocytes, is of positive significance for increasing the proportion of M2 cells, improving the immune microenvironment and providing neuroprotection.

One limitation of this study is that the temporal dynamic analysis and the window period for immune intervention targeting MG only from mouse SCI model. Whether these laws are consistent with human related diseases still need to be further explored.

Conclusions

In summary, this study not only demonstrate that the pro-inflammatory M1 cells mainly come from MG rather than infiltrated Mø after SCI, but also determine their dynamic patterns. Therefore, these findings not only answer the academic debate about which of the infiltrating Mø and MG plays a key role, but also determines the appropriate time window of immune intervention targeting M1-type MG for the treatment of SCI.

Supplemental Information

Supplemental Information 1 Raw data.

Click here for additional data file.

Supplemental Information 2 Author Checklist.

Click here for additional data file.

Additional Information and Declarations

Competing Interests

Author Contributions

Animal Ethics

Data Availability

The authors declare that they have no competing interests.

Jing-Lu Li performed the experiments, prepared figures and/or tables, and approved the final draft.

Gui-Qiang Fu performed the experiments, prepared figures and/or tables, and approved the final draft.

Yang-Yang Wang performed the experiments, prepared figures and/or tables, and approved the final draft.

Ming-Ming Bian performed the experiments, prepared figures and/or tables, and approved the final draft.

Yao-Mei Xu performed the experiments, prepared figures and/or tables, postoperative care, and approved the final draft.

Lin Zhang performed the experiments, prepared figures and/or tables, animal feeding, and approved the final draft.

Yu-Qing Chen analyzed the data, prepared figures and/or tables, and approved the final draft.

Nan Zhang analyzed the data, prepared figures and/or tables, and approved the final draft.

Shu-Qin Ding analyzed the data, authored or reviewed drafts of the article, and approved the final draft.

Rui Wang analyzed the data, authored or reviewed drafts of the article, and approved the final draft.

Rui Fang analyzed the data, authored or reviewed drafts of the article, and approved the final draft.

Jie Tang analyzed the data, authored or reviewed drafts of the article, and approved the final draft.

Jian-Guo Hu conceived and designed the experiments, authored or reviewed drafts of the article, and approved the final draft.

He-Zuo Lü conceived and designed the experiments, authored or reviewed drafts of the article, and approved the final draft.

The following information was supplied relating to ethical approvals (i.e., approving body and any reference numbers):

The animal care and use committee of Bengbu Medical College provided full approval for this research (2017-037).

The following information was supplied regarding data availability:

The study data are available in the Supplemental Files.

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
