# Peer review of "The polarization of microglia and infiltrated macrophages in the injured mice spinal cords: a dynamic analysis"

_PeerJ, doi:10.7717/peerj.14929_

## Round 0.1 · original submission · Major Revisions

The authors need to carefully address the reviewers' comments on methodology and data description and discussion. Also, Introduction section and reference citation need to be improved.

Reviewer 1 ·

Basic reporting

The Result section is verbose and should be more simplified.

Experimental design

No comment.

Validity of the findings

Line 295 “To verify the temporal pattern of MG and infiltrated M¯ detected by FCM, several representative spinal cords (sham, 1, 7 and 28 dpi) were selected for IHF analysis.” Can you please specify how the representative spinal cords were selected. Like was it random selected or based on some method.

Line 47 “received a laminectomy without contusive injury. Fow cytometry and immunohistofluorescence” should be flow cytometry. Same typo in the background.

Line 210. “There were no significant differences among 7, 14, 21 and 28 dpi (P > 0.05, n = 6). However, the percentages of CD68+ cells in these four groups were significant higher comparing with sham (P < 0.01, n = 6), 1 and 3 dpi (P < 0.05, n = 6).” The authors discussed the increase 3dpi. Can you please elaborate why no significant differences among 7, 14, 21 and 28 dpi in Disscussion.

Additional comments

Figure 1 legend. The conclusive figure that explains the pathway and the whole idea of this research. Name it a flow chart should be more appropriate.

Reviewer 2 ·

Basic reporting

This article is written in English clearly.

Suggestion:
Relevant research background needs to be supplemented in INTRODUCTION. Please cite all references used in this paper properly.

Experimental design

no comment.

Validity of the findings

This study found that the total Mø/MG gradually increased and peaked at 7 dpi, and maintained at high levels 14, 21 and 28 dpi.

suggestion:
1) In Abstract Results, " the activated MG increased nearly to 90% " should be changed to " activated MG increased to nearly 90% ".
2) Please check the scales of immunofluorescence images carefully.

---

## Round 0.2 · accepted · Accept

The authors have addressed all of the reviewer's comments and now the manuscript can be accepted for publication.

Reviewer 1 ·

Basic reporting

The authors addressed all reviewer concerns in the revision.

Experimental design

No comment.

Validity of the findings

The authors responded well and addressed all reviewer concerns.

Additional comments

No comments.

Reviewer 2 ·

Basic reporting

no comment.

Experimental design

no comment.

Validity of the findings

no comment.

Additional comments

The authors have revised this paper based on the comments.